# The Role of *Petrimonas mucosa* ING2-E5A^T^ in Mesophilic Biogas Reactor Systems as Deduced from Multiomics Analyses

**DOI:** 10.3390/microorganisms8122024

**Published:** 2020-12-17

**Authors:** Irena Maus, Tom Tubbesing, Daniel Wibberg, Robert Heyer, Julia Hassa, Geizecler Tomazetto, Liren Huang, Boyke Bunk, Cathrin Spröer, Dirk Benndorf, Vladimir Zverlov, Alfred Pühler, Michael Klocke, Alexander Sczyrba, Andreas Schlüter

**Affiliations:** 1Center for Biotechnology (CeBiTec), Genome Research of Industrial Microorganisms, Bielefeld University, Universitätsstr. 27, 33615 Bielefeld, Germany; irena.maus@CeBiTec.Uni-Bielefeld.de (I.M.); dwibberg@cebitec.uni-bielefeld.de (D.W.); jhassa@cebitec.uni-bielefeld.de (J.H.); puehler@cebitec.uni-bielefeld.de (A.P.); 2Faculty of Technology, Bielefeld University, Universitätsstr. 25, 33615 Bielefeld, Germany; t.tubbesing@uni-bielefeld.de (T.T.); huanglr@cebitec.uni-bielefeld.de (L.H.); asczyrba@cebitec.uni-bielefeld.de (A.S.); 3Bioprocess Engineering, Otto von Guericke University Magdeburg, Universitätspl. 2, 39106 Magdeburg, Germany; heyer@mpi-magdeburg.mpg.de (R.H.); benndorf@mpi-magdeburg.mpg.de (D.B.); 4Database and Software Engineering Group, Department of Computer Science, Institute for Technical and Business Information Systems, Otto von Guericke University Magdeburg, Universitätspl. 2, 39106 Magdeburg, Germany; 5Department of Bioengineering, Leibniz Institute for Agricultural Engineering and Bioeconomy, Max-Eyth-Allee 100, 14469 Potsdam, Germany; 6Biological and Chemical Engineering Section (BCE), Department of Engineering, Aarhus University, 8000 Aarhus, Denmark; geizetomazetto@gmail.com; 7Department Bioinformatics and Databases, Leibniz Institute DSMZ-German Collection of Microorganisms and Cell Cultures GmbH, Inhoffenstraße 7B, 38124 Braunschweig, Germany; Boyke.Bunk@dsmz.de (B.B.); ckc@dsmz.de (C.S.); 8Bioprocess Engineering, Max Planck Institute for Dynamics of Complex Technical Systems, Sandtorstr. 1, 39106 Magdeburg, Germany; 9Microbiology, Anhalt University of Applied Sciences, Bernburger Straße 55, 06354 Köthen, Germany; 10Chair of Microbiology, Technical University of Munich, Emil-Ramann-Str. 4, 85354 Freising, Germany; vladimir.zverlov@tum.de; 11Institute of Molecular Genetics, National Research Centre «Kurchatov Institute», Kurchatov Sq. 2, 123128 Moscow, Russia; 12Institute of Agricultural and Urban Ecological Projects Affiliated to Berlin Humboldt University (IASP), Philippstraße 13, 10115 Berlin, Germany; mklocke@posteo.de

**Keywords:** carbohydrate-active enzymes, polysaccharide utilization loci, anaerobic digestion, biomethanation, metabolic pathway reconstruction, bioconversion

## Abstract

Members of the genera *Proteiniphilum* and *Petrimonas* were speculated to represent indicators reflecting process instability within anaerobic digestion (AD) microbiomes. Therefore, *Petrimonas mucosa* ING2-E5A^T^ was isolated from a biogas reactor sample and sequenced on the PacBio *RSII* and Illumina MiSeq sequencers. Phylogenetic classification positioned the strain ING2-E5A^T^ in close proximity to *Fermentimonas* and *Proteiniphilum* species (family Dysgonomonadaceae). ING2-E5A^T^ encodes a number of genes for glycosyl-hydrolyses (GH) which are organized in Polysaccharide Utilization Loci (PUL) comprising tandem *sus*CD-like genes for a TonB-dependent outer-membrane transporter and a cell surface glycan-binding protein. Different GHs encoded in PUL are involved in pectin degradation, reflecting a pronounced specialization of the ING2-E5A^T^ PUL systems regarding the decomposition of this polysaccharide. Genes encoding enzymes participating in amino acids fermentation were also identified. Fragment recruitments with the ING2-E5A^T^ genome as a template and publicly available metagenomes of AD microbiomes revealed that *Petrimonas* species are present in 146 out of 257 datasets supporting their importance in AD microbiomes. Metatranscriptome analyses of AD microbiomes uncovered active sugar and amino acid fermentation pathways for *Petrimonas* species. Likewise, screening of metaproteome datasets demonstrated expression of the *Petrimonas* PUL-specific component SusC providing further evidence that PUL play a central role for the lifestyle of *Petrimonas* species.

## 1. Introduction

Anaerobic digestion (AD) is commonly applied for treatment of organic wastes in order to achieve reduction of waste combined with parallel recovery of bioenergy [1,2]. The degradation of organic matter is a complex microbial process featuring interaction of several groups of microorganisms with different interacting or even interfering metabolic capabilities and growth requirements [3]. The microbial biogas network is complex and intertwined and therefore susceptible to environmental changes. Several factors causing process inhibition have been identified, such as temperature fluctuations, foaming, high concentrations in hydrogen sulfide or ammonia [4,5,6,7], imbalanced levels of light or heavy metals, the presence of interfering organic substances such as halogenated alkanes or aromatic compounds [8] as well as increased organic loading rates (OLR) [9,10]. Among these, the most frequent incident type described in literature resulting in process inhibition is the accumulation of ammonia [3]. Due to inhibition of syntrophic volatile fatty acid (VFA) oxidizers, high ammonia concentrations result in VFA accumulation which subsequently inhibits methanogenesis. Moreover, ammonia also directly inhibits methanogens [11].

In the past decades, remarkable progress has been made towards understanding of how disturbances affect the microbial community structures, biodiversity losses, and ecological processes [12,13,14,15,16,17,18]. Due to further improvements and developments in high-throughput sequencing technologies, deep insights into the microbial community became possible, allowing predictions on interrelations between the microbial community composition, interspecies interactions, and species as well as community dependencies concerning supplied AD substrates and deteriorating process parameters. In connection with the formation of VFA [19] and increased ammonium concentrations, it has been observed that members of the phylum Bacteroidetes, namely species of the families Dysgonomonadaceae (formerly Porphyromonadaceae) and Marinilabiaceae, are frequently abundant in mesophilic biogas reactor systems [20,21]. Often, corresponding biogas plants were fed with food waste or protein-rich and hardly digestible substrates such as straw showing unstable process flow with variations in biogas/methane yields and increased nitrogen/ammonia levels [13,22].

The family Dysgonomonadaceae comprises species that frequently are associated with a variety of human infections [23], but also species detected in different types of biogas reactor systems. The latter are closely related to *Proteiniphilum acetatigenes* [24], *Fermentimonas caenicola* [25], and *Petrimonas sulfuriphila* [26]. Currently, the family Dysgonomonadaceae comprises the genera *Dysgonomonas*, *Fermentimonas*, *Petrimonas*, and *Proteiniphilum* [27]. The genera *Fermentimonas* and *Petrimonas*, originally assigned to the family Porphyromonadaceae [25], were recently transferred to the family Dysgonomonadaceae and now represent a sister group of the genus *Proteiniphilum* [27]. Besides utilization of complex proteinaceous substrates such as yeast extract and peptone, members of the Dysgonomonadaceae family are able to ferment a wide range of mono- and disaccharides [23,25,28,29].

The impact of Dysgonomonadaceae members on biomass degradation in AD reactor systems operating at high OLR or increased nitrogen/ammonia levels caused by proteinaceous substrates has been demonstrated recently [9,10]. Under mesophilic conditions, an increase in OLR had a positive effect on the abundance of Dysgonomonadaceae members of the genera *Proteiniphilum* and *Petrimonas* [9]. These observations led to the assumption that these genera may serve as microbial early warning indicators for an unbalanced process. They are therefore regarded as potential marker microorganisms featuring outstanding performance under stress conditions [9,10]. Moreover, a study published by He and coworkers [7] demonstrated the proliferation of members of the genus *Petrimonas* in biogas reactors that were operated under unfavorable or unstable conditions indicated by foaming. The relative abundance of *Petrimonas* was highest (29.34%) in foaming samples, and this value was almost seven times higher than those for non-foaming samples. Hence, it was speculated that the presence of Dysgonomonadaceae members, in particular of acid-producing *Petrimonas* bacteria, might help to optimize VFA compositions and to accelerate methane production by driving the formation of acetic acid and carbon dioxide [19].

To overcome the limited knowledge on microbial species affecting AD in a positive or negative way, a number of previously unknown bacterial species were isolated from biogas reactor samples and subsequently characterized regarding their physiological and genetic features [25,30,31,32,33,34]. Among these, *Petrimonas mucosa* ING2-E5A^T^ was isolated from a mesophilic laboratory-scale biogas reactor [25]. Microbiological characterization revealed that this strain is able to utilize complex proteinaceous substrates, as well as to ferment mono- and disaccharides. Considering these phenotypic characteristics required for effective biomass conversion, the aim of this study was to establish and analyze the complete genome sequence of *P. mucosa* ING2-E5A^T^ in order to understand the strain’s functioning and competitiveness in biogas plant environments.

## 2. Materials and Methods 

### 2.1. Strain Cultivation and DNA Isolation

*P. mucosa* ING2-E5A^T^ was obtained from a sample taken from a mesophilic laboratory-scale biogas reactor continuously fed with a mixture of maize silage (10% of organic dry matter, ODM), pig (45% ODM) and cattle manure (45% ODM) as described previously [25]. The isolate ING2-E5A^T^ was grown at 37 °C on Columbia Agar Base medium supplemented with 5% horse blood and 1.7 g/L sodium propionate. The extraction of genomic DNA was performed using the Gentra Puregene Yeast/Bact. Kit (Qiagen, Hilden, Germany) following the manufacturer’s instructions. Unfortunately, required DNA purity criteria being A260/280 and A260/230 values greater than 1.8 were not achieved after DNA extraction with the Gentra Puregene Yeast/Bact. Kit. Therefore, obtained DNA was additionally purified using the NucleoSpin^®^ gDNA Clean-up kit (Macherey-Nagel, Düren, Germany).

### 2.2. PacBio Library Preparation, Sequencing, and Genome Assembly

Approximately 8 μg genomic DNA was used for the SMRTbell^TM^ template library construction, which was prepared according to the instructions from Pacific Biosciences (Menlo Park, CA, USA) following the ’Procedure and Checklist—Greater Than 10 kb Template Preparation’. SMRT sequencing was carried out on the PacBio *RSII* (Pacific Biosciences, MenloPark, CA, USA) taking one 240-min movie for two SMRT cells using the P6 Chemistry. Subsequently, SMRT Cell data was assembled using the ‘RS_HGAP_Assembly.3’ protocol included in the SMRT Portal version 2.3.0 using default parameters. The assembly revealed a circular chromosome. The ‘RS_Bridgemapper.1’ protocol was applied to check the validity of the assembly. The chromosome was circularized and particularly artificial redundancies at the ends of the contigs were removed and adjusted to *dna*A as the first gene. Finally, error-correction was performed by a mapping of paired-end MiSeq Illumina reads of 2 × 300 bp obtained previously for the strain ING2-E5AT [34] onto the finished genome using BWA [35] with subsequent variant and consensus calling using VarScan [36]. For further genome assembly details refer to [37].

The finished genome sequence was imported into the annotation platform GenDB [38] for automatic prediction of genes as described previously [29]. To identify phage-related genes and genomic islands (GI), the *P. mucosa* ING2-E5AT genome was uploaded in PHASTER [39] and IslandViewer 3.0 [40]. Finally, the program EDGAR 2.0 [41], a software tool for the comparative analysis of prokaryotic genomes, was applied in order to phylogenetically classify completely sequenced and annotated members of the order *Bacteroidales* as well as to analyze them in a comparative manner.

### 2.3. Reconstruction of Metabolic Pathways

To predict genes encoding carbohydrate-active enzymes, the carbohydrate-active enzyme database (CAZy) annotation web-server dbCAN [42] was applied. Metabolic pathways of interest were reconstructed based on EC numbers of enzymes and their assignment to KEGG pathway maps (https://www.genome.jp/kegg/pathway.html). Hydrogenases were identified using blastp and the database HydDB with a threshold of 1 × 10^−100^. In addition, further classification of the predicted hydrogenases was also performed by means of HydDB [43].

### 2.4. Fragment Recruitment

To determine the distribution and abundance of *Petrimonas mucosa* in different biogas communities, fragment recruitments using the genome sequence of *Petrimonas mucosa* as template were performed by application of the bioinformatics tool SparkHit [44]. Corresponding computations were scaled-up and parallelized by using the de.NBI Cloud environment (https://www.denbi.de/cloud). As a fast and sensitive fragment recruitment tool, the so-called Sparkhit-recruiter was applied. This tool extends the FR-hit pipeline [45] and is implemented natively on top of the Apache Spark. In addition, it integrates a series of analytical tools and methods for various genomic applications. The fragment recruitment option implements the q-Gram algorithm to allow more mismatches than a regular read mapping during the alignment, so that extra information is provided for the metagenomic analysis. SparkHit was applied on metagenome FASTQ files from 257 biogas metagenome datasets that were downloaded from ENA (www.ebi.ac.uk/). Randomly chosen two million reads of each FASTQ file were compared to the *P. mucosa* ING2-E5A^T^ genome. The alignment identity threshold was set to > 50% similarity. The result of the fragment recruitments were visualized using Circos [46].

### 2.5. Petrimonas Transcriptional Profile as Deduced from Public Transcriptome Datasets

Metatranscriptome datasets pertaining to samples from different industrial- and laboratory-scale anaerobic digesters, detailed in Appendix A, were reanalyzed to quantify transcripts of *Petrimonas mucosa*. Raw RNA-sequencing reads were trimmed with Trim Galore [47]. Subsequently, reads were filtered with SortMeRNA [48] to exclude ribosomal RNA from the downstream analysis. RnaSPAdes [49] was used to assemble transcripts from read datasets originating from the same biogas reactor or wastewater treatment plant. Reads from samples representing biological or technical replicates were pooled into sets. For each of these sets, the tool Kallisto [50] was used to assign read counts to the assembled metatranscriptome-contigs in case of paired-end reads. For single-end reads, Bowtie2 [51] was used to map reads to the metatranscriptome-contigs and uniquely mapping reads were counted for each contig. All contigs resulting from the transcriptome assembly were aligned to the genome of *P. mucosa* ING2-E5A^T^ using nucleotide BLAST [52]. Metatranscriptome-contigs for which BLAST produced alignments covering at least 99% of the contig length with an identity of 97% or more were classified as transcripts originating from *P. mucosa*. The previously produced read counts for each of these contigs were assigned to features in the *P. mucosa* genome based on the annotated strain ING2-E5A^T^ genome. Individual contigs may overlap multiple coding sequences of the genome. In such cases, the read counts that were computed for a contig were split up between all coding sequences which the contig is overlapping. A fraction $X_{i}$ of total contig-read-counts X is assigned to each coding sequence i. $X_{i}$ is determined by dividing the length of the sequence $l_{i}$ through the sum of the length of all sequences j which are overlapping the contig:$$X_{i}\;=\;X\;*\;\frac{l_{i}}{\sum_{j}l_{j}}$$

In order to rank *P. mucosa* features based on their expression levels, transcripts per million (TPM) were calculated for each feature. Only transcripts assigned to *P. mucosa* were included in the TPM calculations.

### 2.6. Gene Expression Analysis of Petrimonas Species Based on Database Metaproteome Datasets

High-resolution metaproteomics datasets from ten industrial biogas plants and one laboratory-scale biogas fermenter (PRIDE ACCESSION: PXD009349) [53] were reanalyzed to quantify the abundance of *Petrimonas mucosa* proteins. For each of the biogas plants, the first replicate of the first sampling time point was searched with the database search engines X!Tandem [54] and OMMSA [55] implemented in the MetaProteomeAnalyser [56] (Version 3.0) against the metagenome database from the original study [57,58,59,60] extended by the protein sequences of *P. mucosa*. The remaining settings were used as described previously by Heyer and colleagues [56] with the minor change that redundant homologous proteins were grouped when they shared the same peptide.

### 2.7. Nucleotide Sequence Accession Number

The genome of *P. mucosa* ING2-E5A^T^ was deposited in the EMBL-EBI database (European Bioinformatics Institute database; https://www.ebi.ac.uk/ena/browser/view/LT608328) under the accession number LT608328.

## 3. Results and Discussion

### 3.1. General Features of Petrimonas mucosa ING2-E5A^T^ Genome

To deduce the importance of *Petrimonas* members in biogas-producing microbial communities, the genome of the *P. mucosa* strain ING2-E5A^T^ (= DSM 28695^T^ = CECT 8611^T^) was completely sequenced and analyzed in detail. General features of the *P. mucosa* ING2-E5A^T^ genome are summarized in Table 1 and Figure 1. The sequencing approach yielded 115,545 reads with a mean read length of 8488 bp, accounting for 980,798,552 bp sequence information. Long read genome assembly followed by short read error correction resulted in one circular contig with 3,717,632 bp in size, featuring a GC-content of 47.97%. Gene prediction and annotation of the *P. mucosa* ING2-E5A^T^ genome sequence resulted in the identification of 3000 coding sequences, 49 tRNA genes, and two *rrn* operons. Analysis of the *P. mucosa* ING2-E5A^T^ genome for the presence of horizontally acquired DNA elements resulted in the identification of 27 genomic islands (Figure 1) and putative phage genes. However, no complete prophage cluster was predicted in the genome. ING2-E5A^T^ possesses one CRISPR-*cas* system which may play a role in preventing invasion of phages and mobile genetic elements.

### 3.2. Phylogenetic Classification of Petrimonas mucosa ING2-E5A^T^

Phylogenetic tree reconstruction was performed based on available genomes for 47 strains of the order Bacteroidales comprising 94 core genes in total (Figure 2). The analysis resulted in a distinct cluster of the genus *Petrimonas* together with the genera *Fermentimonas* and *Proteiniphilum*. Originally, these genera were assigned to the order Bacteroidales, and within this order, they were affiliated with to the family Porphyromonadaceae as deduced from 16S rRNA gene analyses [25,61]. However, in the obtained core gene-based phylogenetic tree, which is in accordance with the recent taxonomic revision within the phylum Bacteroidetes [27], *Porphyromonas* species are located in a separated cluster with notable distance to the *Petrimonas*/*Proteiniphilum* cluster which is much closer related to the genus *Dysgonomonas*. Therefore, members of the genera *Petrimonas*, *Proteiniphilum*, *Fermentimonas*, as well as *Dysgonomonas* constitute the current family Dysgonomonadaceae, which was earlier suggested by Ormerod et al. [62] and has only recently been validated [27].

The genus *Petrimonas* was firstly described by Grabowski et al. [26] based on cultivation, classification, and characterization of the type species *P. sulfuriphila*. So far, *P. mucosa* and *P. sulfuriphila* were the only validly described species for this genus. Moreover, within this genus, only for the type strain *P. mucosa* ING2-E5A^T^ [25] the genome sequence is currently available. By 16S rRNA gene sequence analysis and DNA–DNA hybridization, *P. mucosa* ING2-E5A^T^ appeared to be closely related to *P. sulfuriphila* BN3^T^ with 97% 16S rRNA gene sequence identity and a DNA–DNA relatedness of 23.8–25.7% [25].

### 3.3. Genes Encoding Carbohydrate-Active Enzymes

To elucidate the *P. mucosa* ING2-E5A^T^ genes involved in utilization of different carbohydrates and sugar components and to classify the strain’s lifestyle, the carbohydrate-active enzyme database (CAZy) annotation web-server dbCAN [42] was applied. CAZymes are a class of enzymes which synthesize, modify, or break down different carbohydrates. Their classification comprises the following families: glycoside hydrolase families (GH), polysaccharide lyase families (PL), carbohydrate esterase families (CE), glycosyltransferase families (GT), auxiliary activity families (AA), and carbohydrate-binding module (CBM) families [63]. The proportion of CAZymes predicted in the ING2-E5A^T^ genome was calculated to represent 7.6% of all gene products encoded in the genome, indicating the strain’s ability to process various polysaccharides. Of a total of 233 CAZymes encoded in the ING2-E5A^T^ genome, 154 GHs account for 44 different families (Figure 3). Previous studies on Gram-negative bacteria of the phylum Bacteroidetes also reported on a large variety of different GH families in these species (up to 62) [64]. Based on these observations, a broad biodegradation potential regarding carbohydrates is proposed for this phylum. The most dominant GH families in *P. mucosa* ING2-E5A^T^ were the GH109, GH2, GH92, GH127, and GH106 families, listed by decreasing abundance. These families mostly encode glycosidases which hydrolyze single sugar residues from the non-reducing ends of di- and polysaccharides such as β-d-galactose (GH2), arabinosides (GH127), glycan (GH92), α-l-rhamnose (GH106), and α-(1→3)-N-acetylgalactosamine (GH109). Several other GH families identified in the ING2-E5AT genome are also associated with hydrolysis of single sugar residues indicating the importance of these substrates for this strain. Further results obtained for *P. mucosa* ING2-E5A^T^ also showed that it encodes enzymes involved in cell wall degradation such as lysozyme (GH23, GH25, GH73) and chitinases (GH19, GH76). Hence, *P. mucosa* ING2-E5A^T^ most probably can utilize peptidoglycan and chitin for growth. These polysaccharides are usually part of the bacterial cell wall or are found in fungi and insects, respectively, indicating the potential of *P. mucosa* ING2-E5A^T^ for further biotechnological applications. As expected for a member of the phylum Bacteroidetes, no genes encoding cohesin-containing putative scaffoldins and corresponding dockerin-containing glycoside hydrolases featuring a potential for cellulosome formation were identified. However, the genome of the strain ING2-E5A^T^ encodes enzymes predicted to degrade cellulose (GH5) and hemicellulose (GH16, GH28, GH53, and GH141).

### 3.4. Genes Encoding Polysaccharide Utilization Loci (PUL)

Degradation of the most abundant plant polysaccharide (cellulose) has usually been affiliated to the metabolic capabilities of members of the phylum Firmicutes [65,66]. According to the current literature, the Bacteroidetes are not associated with cellulose degradation. However, various members of this phylum encode a large diversity of polysaccharide utilization loci (PUL) which were suggested to be in combination with enzymes of the GH5 family acting on cellulose, as an alternative system for cellulose degradation in Bacteroidetes members, in addition to cellulosomes and free-enzyme systems [67,68]. PUL comprise clusters of genes encoding surface glycan binding proteins, transporters, and a variety of glycan-degrading enzymes. The genetic architecture is characterized by tandem *sus*CD-like genes encoding a TonB-dependent outer-membrane transporter (TBDT) and an N-terminally lipidated cell surface glycan-binding protein (SGBP), respectively [68].

The *P. mucosa* ING2-E5A^T^ genome contains 39 PUL (Figure 4), 25 of which comprise 62 GHs belonging to 24 distinct CAZyme families, representing 25.1% of all families predicted in the genome. The predominant families are GH28 and GH92 representing enzymes involved in pectin and glycan degradation (Figure 4). Besides GH28, other families related to pectin degradation are also associated with PUL, such as GH2, GH43, GH78, GH105, GH106, CE8, CE12, and PL1. Accordingly, 29.9% of all CAZymes located in PUL are involved in pectin degradation, reflecting a pronounced specialization of the enzymatic system for the decomposition of this polysaccharide. Other PUL were predicted to be involved in degradation of arabinan (e.g., PUL_12), galactan (PUL_39), and hemi-/cellulose (PUL_11, PUL_19).

In addition to the genes for CAZymes, the *P. mucosa* ING2-E5A^T^ PUL also are associated with peptidase genes. PUL_05 and PUL_06 (Figure 4) encode peptidases belonging to family M60 and S15, respectively. These enzymes were predicted to target glycoproteins [69] and serine peptidases [70]. Presence of peptidase genes in PUL suggests that strain ING2-E5A^T^ may also be involved in protein degradation and subsequent amino acid fermentation [29,71]. However, it was recently reported that the strain ING2-E5A^T^ was not able to grow on peptone or tryptone as sole carbon sources [25]. It may be speculated that growth on these substrates requires additional micronutrients that were not provided in the growth experiments conducted previously [25].

Approximately 46% of the *P. mucosa* PUL are associated with regulator system genes, for example genes encoding an extracytoplasmic function sigma factor (ECF-σ), an anti-sigma (*anti*-*σ*) factor, and a hybrid two-component system (HTCS). Involvement of corresponding regulators in controlling expression or PUL-associated genes remains to be determined [68]. Association of PUL with genes encoding ECF-σ, anti-σ pairs, HTCS, transcriptional regulators (e.g., LacI, AraC, GnTR), or sensor-regulator systems (SusR) is not unusual and has been reported for PUL of other Bacteroidetes species [68].

Similar characteristics were also observed in the CAZymes profile of *Petrimonas* sp. IBARAKI whose genome sequence is closely related to the one of strain ING2-E5A^T^ (PULD database; [72]. Strain IBARAKI harbors 240 CAZymes, 33% of which are located in PUL. Approximately, 40% of its PUL encode enzymes assigned to CAZyme families involved in pectin degradation. Although there are only two genome sequences of *Petrimonas* species available, it can be assumed that this genus preferably degrades pectin.

In summary, genome sequencing of new Bacteroidetes species may elucidate how members of this phylum adapt PUL for decomposition of particular polysaccharides. Moreover, PUL-encoded enzymes provide exploitation possibilities for biotechnological purposes.

### 3.5. Genes Encoding Enzymes of the Central Fermentation Metabolism

Regarding its central metabolism, *P. mucosa* ING2-E5A encodes complete KEGG modules for glycolysis (Embden–Meyerhof pathway, M00001), pyruvate oxidation (M00307), the citrate cycle (TCA cycle, M00009), and the pentose phosphate pathway (M00006) allowing metabolism of glucose, pentoses, and other monosaccharides. As described above, several di-, oligo-, and polysaccharides potentially can be metabolized to the aforementioned common monosaccharides. *P. mucosa* was also predicted to perform degradation of proteins, oligo- and dipeptides since corresponding enzymatic activities are encoded in its genome. Amino acids resulting from peptide hydrolysis may feed fermentation pathways. For example, enzymes for conversion of l-asparagine via aspartate to fumarate are encoded in the *P. mucosa* genome (see Table 2). Citrate cycle enzymes catalyze the reactions from fumarate via malate to oxaloacetate which can then be decarboxylated to pyruvate by oxaloacetate decarboxylase (EC 4.1.1.112). Subsequently, ATP is generated via the phosphate-acetyltransferase/acetate kinase module (KEGG M00579) that is encoded in the *P. mucosa* genome.

Likewise, glutamine and glutamate may be metabolized to yield succinyl-CoA via 2-oxoglutarate (refer to Table 2). *P. mucosa* encodes the key enzymes methylmalonyl-CoA mutase, methylmalonyl-CoA epimerase and propionyl-CoA carboxylase catalyzing the reactions of succinyl-CoA to (*R*)-methylmalonyl-CoA and subsequently to propionyl-CoA, respectively. The phosphate-acetyltransferase/acetate kinase module (KEGG M00579) is also able to convert propionyl-CoA to propionate via propionyl-phosphate yielding ATP through substrate-level phosphorylation. According to ExPASy (https://enzyme.expasy.org), acetate kinase (EC 2.7.2.1) can also use propanoate as a substrate. Acetate can be converted into the intermediate acetyl-CoA by coupling the reaction of acetate kinase with phosphate acetyltransferase (EC 2.3.1.8) activity. Both, this enzyme and acetate kinase also are important in the production of propanoate. Acetate, propionate, and carbon dioxide were previously identified as fermentation end-products in *P. mucosa* ING2-E5A^T^ [25]. As well as the amino acids mentioned above, other amino acids such as proline, glycine, serine, and threonine potentially can be catabolized via glutamate or pyruvate since the corresponding enzymes are encoded in the *P. mucosa* genome.

Genes for the butanoate pathway enzymes butyrate kinase (EC 2.7.2.7) and phosphate butyryltransferase (EC 2.3.1.19) are present in the *P. mucosa* genome but the pathway is incomplete at least as predicted from the genome sequence information. Genes for the enzymes involved in conversion of 3-hydroxybutanoyl-CoA to crotonoyl-CoA (or *vice versa*) and crotonoyl-CoA to butanoyl-CoA could not be identified in the *P. mucosa* ING2-E5A^T^ genome. However, it was shown that *P. mucosa* ING2-E5A^T^ produced butyric acid in trace amounts when grown on glucose [25]. So far, not all enzymes necessary for butyric acid formation were identified in *P. mucosa*.

Very recently, importance of the phosphotransacetylase/acetate kinase pathway for ATP production was also described in *Porphyromonas gingivalis* which, similar to *P. mucosa*, belongs to the order Bacteroidales [73,74,75,76]. In the context of the central fermentation metabolism, it is interesting to note that *P. mucosa* encodes a malic enzyme (EC 1.1.1.40) and pyruvate, phosphate dikinase (EC 2.7.9.1) catalyzing the NAD(P)^+^ dependent oxidative decarboxylation of malate to pyruvate [77] and the ATP-consuming conversion of pyruvate to phosphoenolpyruvate, respectively. Mostly, the reaction of malic enzyme contributes to NADPH-generation but under specific conditions, it may also function as anaplerotic node by maintaining suitable levels of TCA cycle intermediates [77]. For *P. mucosa*, the biological importance of the reaction catalyzed by malic enzyme remains to be determined.

Furthermore, two hydrogenase gene regions were identified in the *P. mucosa* ING2-E5A^T^ genome by means of BLASTp analyses and consulting the Hydrogenase Database HydDB [43]. The first hydrogenase, being classified as a HydA (ING2E5A_0849), belongs to the [FeFe] Group C1; it is histidine kinase-linked and oxygen labile and its role remains unclear so far. The enzyme may serve as a hydrogen sensor in regulatory cascades. The second hydrogenase Hnd consists of four subunits (HndA, HndB, HndC, and HndD; ING2E5A_0853-0856). The *hnd* genes are located in close proximity to *hydA*. A previous study on the Hnd hydrogenase isolated from *Desulfovibrio fructosovorans* described the electron-bifurcating function of this enzyme coupling an exergonic redox reaction to an endergonic one allowing energy conservation in anaerobic microorganisms [78].

### 3.6. The Occurrence of Petrimonas Relatives in Biogas-Producing Microbial Communities as Deduced from Publicly Available Metagenome Data

To investigate the prevalence and occurrence of *P. mucosa* in biogas-producing microbial communities, metagenome sequences obtained from 257 publicly available datasets originating from the RefSeq collection were mapped onto the ING2-E5A^T^ genome applying the program Sparkhit [44]. The mapping approach was aimed at the reconstruction of *Petrimonas* genomes from the previously published metagenome data. A further objective of this approach was to determine the degree of relatedness of species within the analyzed biogas plant microbiomes to the genome of strain ING2-E5A^T^. Fragment recruitments were done by mapping of at maximum ten million sequences from the corresponding metagenome representing the biogas microbial community. Only five samples (accession numbers at the NCBI: ERR3656077, ERR3654114, SRR8925757, ERR3654113, SRR5351547) showing metagenome fragment mapping results of greater than 1% were visualized in the fragment recruitment plot (Figure 5). The complete report is provided as Appendix A.

The genus *Petrimonas* was detected in 146 out of 257 metagenomes analyzed (≥0.1% of all metagenome sequences) with different abundance values. Particularly high abundance values ranging between 1.03% and 3.16% of all metagenome sequences were detected in 14 out of 257 different biogas reactor metagenomes (Figure 5). In these metagenomes, the genome of *P. mucosa* ING2-E5A^T^ was not only abundant, but also fully covered. The highest number of mapped sequences onto the genome of *P. mucosa* (3.16% of all sequences) was found for the microbiome originating from a production-scale biogas plant located in Germany (Ger-06), operated under mesophilic (40 °C) temperature regime and fed with manure (48%), maize (30%), dung (17%), and grain corn (7%) [79]. Brandt and colleagues [79] also described the widespread occurrence of the genus *Petrimonas* in microbiomes of biogas reactors analyzed within their study. They detected *Petrimonas* spp. in 13 out of 20 German and Swedish biogas plants (BGPs) and presented abundance values of up to 10% (Ger-06) of the entire community.

*Petrimonas* members were also detected in microbiomes originating from a pilot-scale dry anaerobic digester fed with residual biomass from the bovine husbandry (64.2% manure, 18.2% oxidation lagoon water, 3% fresh rumen fluid, 9.6% wood chips, 4.0% corn stover, and 0.9% dust mill) and operated under mesophilic conditions [80]. Interestingly, the highest number of sequences assigned to the *P. mucosa* genome in this reactor (2.87% of all metagenome sequences) was detected a short time after a process disturbance that occurred due to a pump failure during leachate recirculation. Most probably, *P. mucosa* was able to cope with corresponding stress conditions and displaced other competing bacteria that are less stress resistant. This is in line with previous studies that link the genus *Petrimonas* with process disorders [7,9]. Our mapping results also showed an abundance decrease of the genus *Petrimonas* in reactor samples taken after the process disorder (from 1.41% to 1.03%), indicating suppression by other competitive bacteria in the stabilization phase of the biogas process.

Information obtained from the fragment recruitment approach leads to the assumption that *Petrimonas* and its relatives are widespread in biogas plants. *Petrimonas* members probably proliferate when reactor conditions become unstable and the community undergoes a structural change. Under these conditions, they may play an important role in conversion of lignocellulosic biomass and also proteins for subsequent methane production.

### 3.7. The Transcriptional Profile of Petrimonas Species

Transcriptome sequence data reflect the transcriptional activity of the species of interest in their respective habitats. To study the *Petrimonas* transcriptional activity in microbiomes in which this species was found with a relative abundance of above 1% (for more details see above), corresponding metatranscriptome data were searched in the publicly available sequence databases. Unfortunately, no metatranscriptome data exist for the metagenomes presented in Figure 5 referring to the fragment recruitment analyses (see above). To gain insights into the transcriptional profile of *Petrimonas* members and their relatives and to understand their role in AD in general, public databases were searched for metatranscriptome datasets representing AD environments. Not all of these datasets also comprise metagenome data. For metagenome data for which also metatranscriptome data was available, only fractions of the *P. mucosa* genome could be assembled from the DNA reads. For this reason, a solely RNA-based analysis was carried out. Transcriptome assemblies were conducted for communities from 13 different biogas and wastewater treatment plants. Sequence reads of the corresponding studies were grouped into 29 datasets, each representing a distinct condition or sampling time point in one of the analyzed reactors. For each of these 29 datasets, reads were mapped back to the assembled transcripts. The ING2-E5A^T^ genome was used to assign transcripts that originate from *P. mucosa*. Based on the number of reads that could be mapped to each *P. mucosa* transcript, counts and TPM values were determined for each dataset. Detailed information on process parameters and used substrates recorded for the analyzed biogas reactors were summarized in Appendix A as far as corresponding metadata were documented in the literature.

Transcripts assigned to *Petrimonas* and its relatives were detected in 20 out of 29 metatranscriptome datasets analyzed. The highest number of transcripts (0.23–0.53% of all sequences) that could be mapped onto the ING2-E5A^T^ genome was observed for a mesophilic laboratory-scale biogas plant fed with maize silage and separated digestate from other mesophilic full-scale research biogas plants [81] (Appendix A). Grohmann and coworkers [81] reported that the analyzed biogas plant suffered from a process disturbance leading to a strong accumulation of VFA. Acetic and propionic acid concentrations exceeded critical values and were on an alarming level. During the experiment, the reactor was continuously sampled. *Petrimonas* transcripts were consistently detected in all metatranscriptome datasets belonging to the latter study (on average 0.31% of all metatranscriptome sequences taking into account all metagenomes from this study). However, the highest number of transcripts (0.53%) were observed for the sample that was taken when the process had nearly stabilized.

Within the list of highly transcribed genes (Table 3 and Appendix A), those encoding proteins associated with hydrolysis of 6-phosphogluconolactone to 6-phosphogluconic acid via the pentose phosphate pathway, such as 6-phosphogluconolactonase Pgl (EC: 3.1.1.31), were observed. Genes involved in oligopeptide utilization such as the ATP-dependent protease *clp* (EC: 3.4.21.-) and the peptidase IV *spp*A gene (EC: 3.4.21.-) were also transcribed. Furthermore, *P. mucosa* transcribed the *msc*L gene encoding the large-conductance mechanosensitive channel. This pore-forming membrane protein is an important component of the cells’ management during osmotic stress [82]. Moreover, these nanopores can be utilized for the passage of membrane-impermeable compounds or peptides [83]. The latter molecules were predicted to represent fermentation substrates of *Petrimonas* species. Furthermore, *P. mucosa* transcribed the *sus*C gene encoding the TonB-dependent outer-membrane transporter, which is part of PUL clusters presented in Figure 4. Further copies of this gene were also highly transcribed. Transcripts encoding proteins involved in mandatory functions (basic house-keeping functions) such as transcription and translation were not further considered.

Based on genome interpretation and transcriptional profiling, a lifestyle based on carbohydrate decomposition and oligopeptide fermentation is proposed for *P. mucosa* ING2-E5A^T^. Furthermore, results obtained from metatranscriptome analyses are in line with previous findings showing increased abundance of *Petrimonas* members in reactors operating under stressed conditions followed by accumulation of VFA [10,19]. Results obtained within this study also indicated that *Petrimonas* members are competent in fermenting sugars and amino acids, with acetate and propionic acid as the major fermentation products.

### 3.8. Metaproteome Analysis Revealed the Presence of Petrimonas spp. Proteins in Various Biogas-Producing Communities

Metaproteomics became a powerful technique to study the biological functions of microbial communities, to analyze and correlate the taxonomic and functional profiles of microbiomes [84] and also to evaluate the responses of microbial taxa to environmental changes [85,86]. In order to evaluate which proteins/enzymes were expressed by *P. mucosa*, metaproteome data originating from mesophilic and thermophilic biogas reactors (previously published by [53]) were screened for proteins/enzymes that were assigned to this taxon. Further details on operational and process parameters of the investigated BGPs as well as fed substrates are summarized in Figure 6. In total, 11 biogas microbiomes were investigated using SDS-PAGE for prefractionation of proteins and subsequent liquid chromatography (LC) coupled to a high-resolution Orbitrap Elite tandem mass spectrometer (MS/MS). Spectra assigned to *P. mucosa* were identified using the MetaProteomeAnalyzer software [87]. The available datasets were searched with X!tandem and OMSSA against the same protein database complemented by the protein sequences deduced from the *Petrimonas mucosa* genome sequence.

Based on abundances of spectral counts, peptides originating from *P. mucosa* proteins after tryptic digestion were detected in all biogas samples analyzed, showing abundance values in the range of 0.2–1.4% of all spectra identified. The highest number of spectra assigned to *P. mucosa* was detected in BGP_03_T1 (1335) representing a mesophilic plug-flow biogas reactor operating at stable process conditions as deduced from the process parameters recorded for this reactor. Among the highest expressed proteins are many of those annotated as house-keeping proteins e.g., uncharacterized outer membrane proteins, and ribosomal proteins of the 30S and 50S ribosomal subunits. Furthermore, several copies of the TonB-dependent outer-membrane transporters (SusC) were found among the most abundant spectra (all are part of different PUL clusters presented in Figure 4: PUL_02, PUL_10, PUL_11, PUL_24, PUL_29, PUL_31) and represent an essential component of *P. mucosa*’s PUL required for the saccharification of complex carbohydrates. This finding indicates the importance of PUL-encoded enzymes for *P. mucosa*’s lifestyle and provides evidence that corresponding enzymes play a central role for this species.

## 4. Conclusions

Integrated omics analyses are useful for a detailed characterization of both, entire microbiomes and individual microbial species enabling new ways to access the microbiological basis for AD. In this study, the genome sequence of *P. mucosa* ING2-E5A^T^ isolated from a mesophilic laboratory-scale reactor was analyzed in detail including genome-based metabolic reconstruction. In addition, a polyphasic approach exploiting publicly available metagenome, metatranscriptome, and metaproteome datasets complemented the genome interpretation of the strain. Genome-based phylogenetic classification of *P. mucosa* ING2-E5A^T^ positioned this strain in close proximity to *Fermentimonas* and *Proteiniphilum* species belonging to the family Dysgonomonadaceae which is in accordance with the recent taxonomic reclassification within the phylum Bacteroidetes. *Dysgonomonas* species are located in a sister group of the *Petrimonas/Fermentimonas/Proteiniphilum* cluster whereas *Porphyromonas* species are more distantly related to the Dysgonomonadaceae members.

Insights into the strain’s genetic repertoire and metabolic potential revealed that *P. mucosa* ING2-E5A^T^ encodes a diverse set of glycosyl-hydrolyses (GH) involved in carbohydrate metabolism as well as pectin degradation. Most of the corresponding genes are organized in Polysaccharide Utilization Loci (PUL) also comprising tandem *sus*CD-like genes for a TonB-dependent outer-membrane transporter and a cell surface glycan-binding protein. Furthermore, ING2-E5A^T^ possesses genetic determinants needed to utilize amino acids as carbon sources. Its function in the AD process most probably is associated with acidogenesis, which is supported by reconstruction of corresponding metabolic pathways leading to production of acetate, propionate, and carbon dioxide as fermentation end-products.

Finally, publicly available metagenome, metatranscriptome, and metaproteome data were used to follow up the question on spreading of *Petrimonas* and its relatives in different biogas plants and subsequently to study its biological functions in the corresponding environment. The genus *Petrimonas* was prevalent in approximately half of the datasets investigated, supporting the importance of this genus for the AD process. Moreover, fragment recruitment results linked the genus *Petrimonas* with process disorders. Most probably, *P. mucosa* was able to cope with corresponding stress conditions and displaced other competing bacteria that are less stress resistant. Functional profiles of *Petrimonas* members shed new light on the importance of PUL-encoded enzymes for *P. mucosa*’s lifestyle, leading to the evidence that corresponding proteins are indispensable for this species.

In the context of biogas production, an integrated -omics approach is believed to provide deeper insights into functional roles of AD community members, thus facilitating the implementation of new optimization strategies for the biogas process. The knowledge gained from multi-omics analyses will provide the basis for application of the strain in demanding biotechnological processes. *P. mucosa* may also be used as indicator species for unbalanced AD processes in the future. However, further information on the performance of the strain in different AD microbiomes exposed to unfavorable process conditions is needed to precisely determine its functional behavior.

## Figures and Tables

**Figure 1 microorganisms-08-02024-f001:**
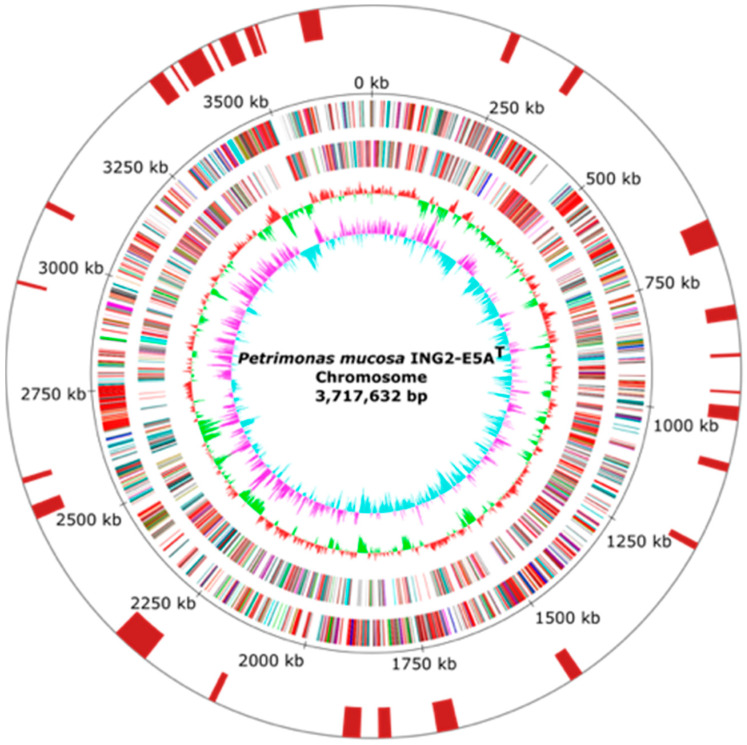
Circular representations of the *P. mucosa* ING2-E5A^T^ chromosome. From the inner to the outer circle: Circle 1: GC skew; Circle 2: GC content; Circles 3 and 4: predicted protein-coding sequences (CDS) transcribed clockwise (outer part) or anticlockwise (inner part). The CDSs are colored according to the assigned Clusters of Orthologous Groups (COG) classes. Circle 5: genomic position in kb. Circle 6: predicted genomic islands (GI) in the ING2-E5A^T^ genome. The replication initiation gene *dna*A was chosen as the first gene of the chromosome.

**Figure 2 microorganisms-08-02024-f002:**
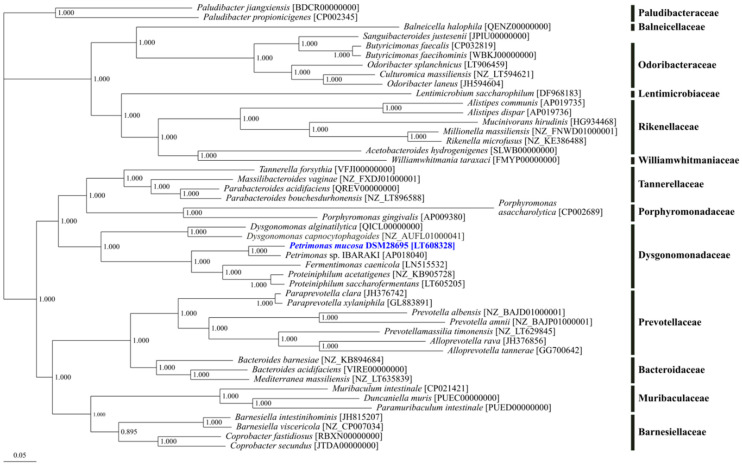
Phylogenetic tree of the order Bacteroidales. The phylogenetic tree is based on 94 core genes of the selected strains as determined by means of the comparative genomics tool EDGAR 2.0 [41]. Except for the metagenome assembled genome *Petrimonas* sp. IBARAKI [AP018040], genome datasets for type strains were used.

**Figure 3 microorganisms-08-02024-f003:**
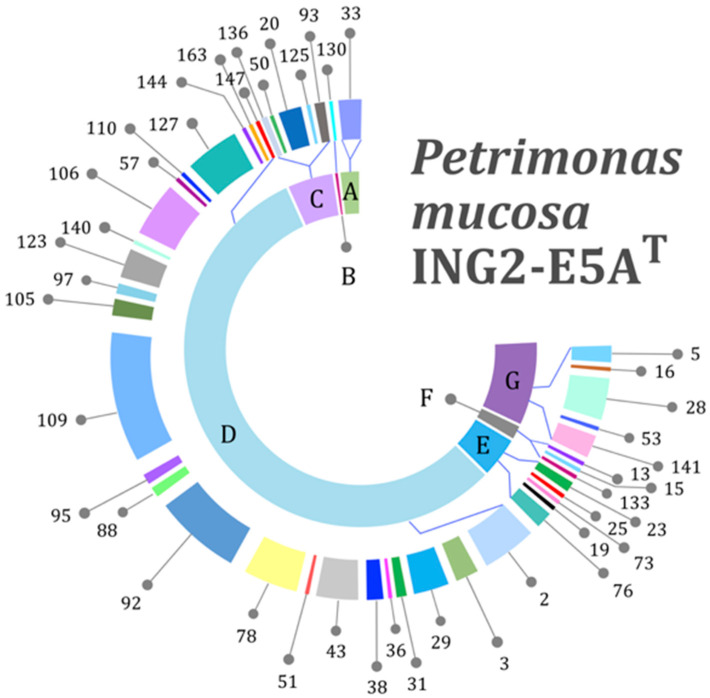
*P. mucosa* ING2-E5A^T^ genes encoding for different glycoside hydrolases (GH) predicted to be involved in carbohydrate utilization as checked manually with the Carbohydrate Active Enzymes (CAZy) database (http://www.cazy.org/). The outer ring represents the diversity of the GH families identified in the genome of the strain ING2-E5A^T^. The inner ring indicates the predicted function of the respective GH family as follows: A: Exo-α-sialidases, B: oligosaccharide phosphorylase, C: glycosidases (hydrolysis of single sugar residues), D: glycosidases (hydrolysis of single sugar residues from nonreducing ends), E: cell wall degradation, F: starch and glycogen hydrolases, G: heteropolymers and polysaccharides including hemicellulose.

**Figure 4 microorganisms-08-02024-f004:**
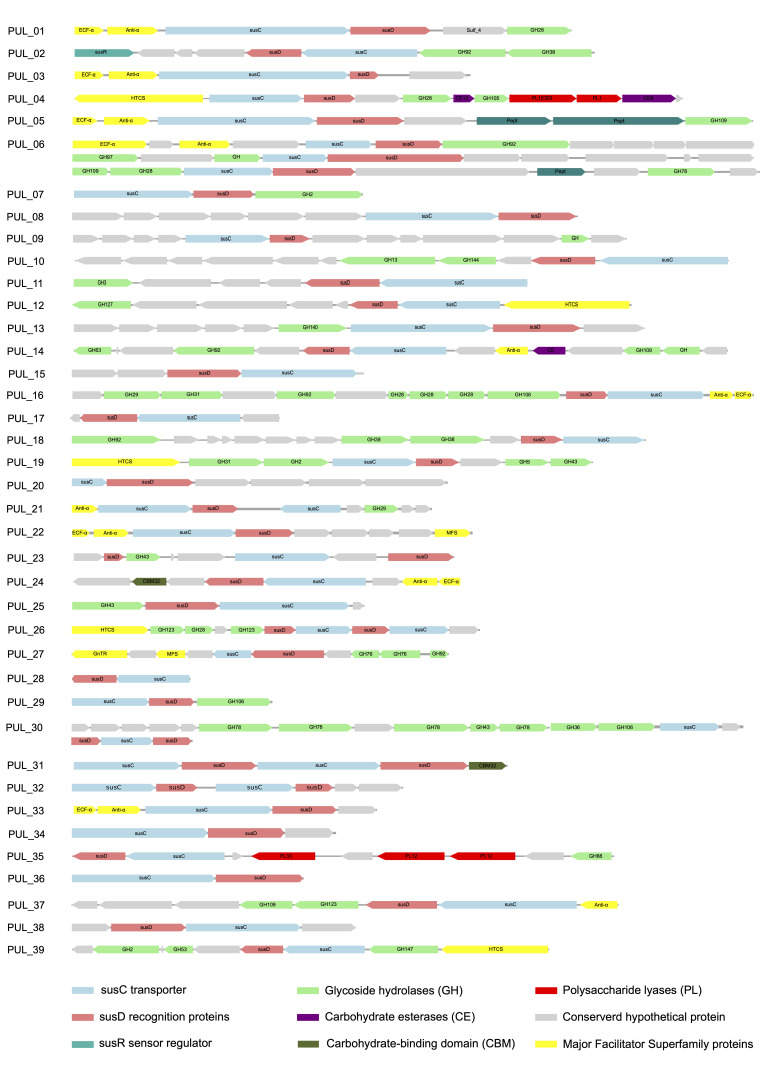
Schematic overview of PUL (Polysaccharide Utilization Loci) predicted in the *P. mucosa* ING2-E5A^T^ genome.

**Figure 5 microorganisms-08-02024-f005:**
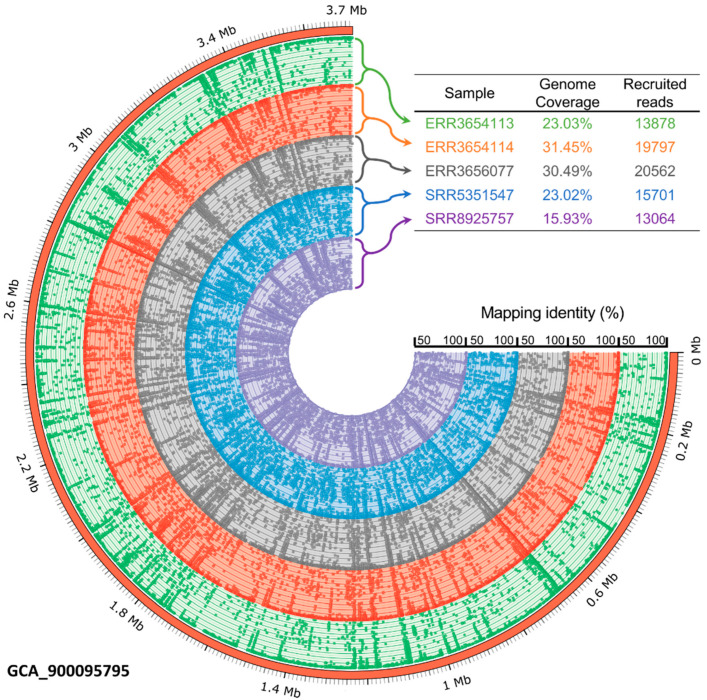
Fragment recruitment of publicly available metagenome sequences derived from five different mesophilic biogas-producing microbial communities and mapped onto the *P. mucosa* ING2-E5A^T^ genome sequence. Selected samples showing metagenome fragment mapping results of greater than 1% were visualized. The outer ring represents the *P. mucosa* ING2-E5A^T^ genome in Mega base pairs (Mb), while remaining colored rings show the coverage of the mapped metagenome reads on the ING2-E5A^T^ genome sequence. Abbreviations: GCA, genome collections accession; ERR and SRR stand for run accessions.

**Figure 6 microorganisms-08-02024-f006:**
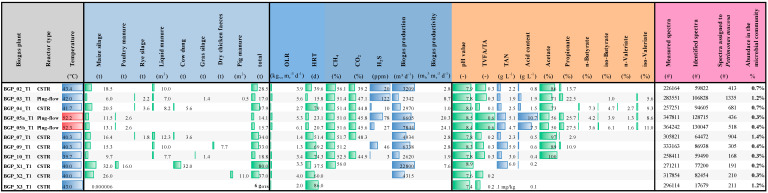
Abundance of *P. mucosa* ING2-E5A^T^ in biogas-producing communities as deduced from metaproteome analysis. Metaproteome datasets originating from eight different biogas plants represented by 11 reactors published previously [53] were investigated. The measurements were searched with X!tandem and OMSSA against the protein database derived from original data extended with the sequences of *Petrimonas mucosa* proteins. Subsequently, an FDR of 1% was applied and redundant peptide identifications were grouped when they shared the same peptide set. The main process parameters for the investigated biogas plants were adopted from the original publication [53]. Blue and red bars mark the mesophilic or thermophilic process temperature, respectively. Green bars visualize the values of the different parameters. CSTR: continuous stirred tank reactor; OLR: organic loading rate; HRT: hydraulic retention time; VS: volatile solids; TS: total solids; TVFA/TA: total volatile fatty acids to total alkalinity; TAN: total ammonia nitrogen; m_b_^3^: cubic meter biogas; m_r_^3^: cubic meter reactor volume; kg_vs_: kilogram VS.

**Table 1 microorganisms-08-02024-t001:** General features of the *P**etrimonas mucosa* ING2-E5A^T^ genome.

Feature	Chromosome
Genome size (bp)	3,717,632
GC content (%)	47.97
Total genes	3055
Protein coding genes	3000
*rrn* operons	2
tRNA genes	49

**Table 2 microorganisms-08-02024-t002:** Catabolism of amino acids and formation of acetate and propionate in *P. mucosa* ING2-E5A^T^.

Locus Tag	EC No.	Enzyme	Reaction	Gene(s)
**Catabolism of Glutamine and Glutamate**				
ING2E5A_1298	3.5.1.2	Glutaminase	l-glutamine + H_2_O <=> l-glutamate + NH_3_	*gls*A
ING2E5A_2384	1.4.1.13	Glutamate synthase (NADPH)	2 l-glutamate + NADP( + ) <=> l-glutamine + 2-oxoglutarate + NADPH	*glt*D
ING2E5A_0283, ING2E5A_2397	1.4.1.3, 1.4.1.2	Glutamate dehydrogenase (NAD(P) ^+^), Glutamate dehydrogenase	l-glutamate + H_2_O + NAD(P)^+^ <=> 2-oxoglutarate + NH_3_ + NAD(P)H, l-glutamate + H_2_O + NAD^+^ <=> 2-oxoglutarate + NH_3_ + NADH	*gdh*A *gdh*
ING2E5A_2385	1.4.7.1	Ferredoxin-dependent glutamate synthase 1	2 l-glutamate + 2 oxidized ferredoxin <=> l-glutamine + 2-oxoglutarate + 2 reduced ferredoxin + 2 H^+^	*glt*B
ING2E5A_1056, ING2E5A_1057	1.2.7.3	2-oxoglutarate synthase	2-oxoglutarate + CoA + 2 oxidized ferredoxin <=> succinyl-CoA + CO_2_ + 2 reduced ferredoxin + 2 H^+^	*kor*B, *kor*A
ING2E5A_1055, ING2E5A_1054	1.2.4.2, 2.3.1.61	Oxoglutarate dehydrogenase (succinyl-transferring), Dihydrolipoyllysine-residue succinyltransferase	2-oxoglutarate → succinyl-CoA	*suc*A, *suc*B
ING2E5A_1515, ING2E5A_1517	5.4.99.2	Methylmalonyl-CoA mutase	(*R*)-methylmalonyl-CoA <=> succinyl-CoA	*mut*A, *mut*B
ING2E5A_1046	5.1.99.1	Methylmalonyl-CoA epimerase	(*R*)-methylmalonyl-CoA <=> (*S*)-2-Methylmalonyl-CoA	
ING2E5A_1045, ING2E5A_1864	6.4.1.3	Propionyl-CoA carboxylase	ATP + propanoyl-CoA + HCO_3_^-^ <=> ADP + phosphate + (*S*)-methylmalonyl-CoA	*pcc*B1 *pcc*B3
ING2E5A_0730	2.3.1.8	Phosphate acetyltransferase	Acetyl-CoA + phosphate <=> CoA + acetyl phosphate	*pta*
ING2E5A_0728	2.7.2.1	Acetate kinase	ATP + acetate <=> ADP + acetyl phosphate	*ack*A
**Catabolism of Arginine and Aspartate**				
ING2E5A_2671	3.5.1.1	l-asparaginase 1	l-asparagine + H_2_O <=> l-aspartate + NH_3_	*ans*A
ING2E5A_2287	4.3.1.1	Aspartate ammonia-lyase	l-aspartate <=> fumarate + NH_3_	*asp*A
ING2E5A_3067	1.4.3.16	l-aspartate oxidase	l-aspartate + O_2_ <=> iminosuccinate + H_2_O_2_	*nad*B
ING2E5A_1308	2.6.1.1	Aspartate transaminase	l-aspartate + 2-oxoglutarate <=> oxaloacetate + l-glutamate	*asp*C
ING2E5A_2589	3.5.3.6	Arginine deiminase	l-arginine + H_2_O <=> l-citrulline + NH_3_	*arc*A
ING2E5A_0737	6.3.4.5	Argininosuccinate synthase	ATP + l-citrulline + l-aspartate <=> AMP + diphosphate + N(omega)-(l-arginino)succinate	*arg*G
ING2E5A_0731	4.3.2.1	Argininosuccinate lyase	2-(N(omega)-l-arginino)succinate <=> fumarate + l-arginine	*arg*H
ING2E5A_3031, ING2E5A_3032, ING2E5A_3033	1.3.5.1	Succinate dehydrogenase, Succinate dehydrogenase flavoprotein subunit	Succinate + a quinone <=> fumarate + a quinol	*sdh*A
ING2E5A_0952, ING2E5A_0953	6.2.1.5	Succinate-CoA ligase (ADP-forming)	ATP + succinate + CoA <=> ADP + phosphate + succinyl-CoA	*suc*D, *suc*C
ING2E5A_0662	4.2.1.2	Fumarate hydratase	(*S*)-malate <=> fumarate + H_2_O	*fum*A
ING2E5A_2725	1.1.1.37	Malate dehydrogenase	(*S*)-malate + NAD^+^ <=> oxaloacetate + NADH	*mdh*
ING2E5A_1041, ING2E5A_1042	4.1.1.112	Oxaloacetate decarboxylase	Oxaloacetate <=> pyruvate + CO_2_	*oad*B2-1, *oad*B2-2
ING2E5A_1825	4.1.1.112	Oxaloacetate decarboxylase	Oxaloacetate <=> pyruvate + CO_2_	*oad*B
ING2E5A_0270, ING2E5A_1826	6.4.1.1	Pyruvate carboxylase	ATP + pyruvate + HCO_3_^-^ <=> ADP + phosphate + oxaloacetate	*pyc*B1, *pyc*B3
ING2E5A_1490, ING2E5A_1805, ING2E5A_1806,	1.2.4.1	Pyruvate dehydrogenase	Pyruvate + [dihydrolipoyllysine-residue acetyltransferase] lipoyllysine <=> [dihydrolipoyllysine-residue acetyltransferase] S-acetyldihydrolipoyllysine + CO(2)	*pdh*B1, *pdh*B3, *pdh*A
ING2E5A_1158, ING2E5A_1264, ING2E5A_1491, ING2E5A_1804	2.3.1.12	Dihydrolipoyllysine-residue acetyltransferase	Acetyl-CoA + enzyme N(6)-(dihydrolipoyl)lysine <=> CoA + enzyme N(6)-(S-acetyldihydrolipoyl)lysine	*pdh*C1, *dla*T, *pdh*C3, *pdh*C5
ING2E5A_0730	2.3.1.8	Phosphate acetyltransferase	Acetyl-CoA + phosphate <=> CoA + acetyl phosphate	*pta*
ING2E5A_0728	2.7.2.1	Acetate kinase	ATP + acetate <=> ADP + acetyl phosphate	*ack*A
**Butanoate pathway enzymes**				
ING2E5A_0177	2.7.2.7	Butyrate kinase	ATP + butanoate <=> ADP + butanoyl phosphate	*buk*
ING2E5A_0176	2.3.1.19	Phosphate butyryltransferase	Butanoyl-CoA + phosphate <=> CoA + butanoylphosphate	*ptb*
ING2E5A_0729	1.1.1.157	3-hydroxybutyryl-CoA dehydrogenase	(*S*)-3-hydroxybutanoyl-CoA + NADP^+^ <=> 3-acetoacetyl-CoA + NADPH	*hbd*
**Malic Enzyme Module**				
ING2E5A_2598	1.1.1.40	Malate dehydrogenase (oxaloacetate-decarboxylating) (NADP^+^).	(*S*)-malate + NADP^+^ <=> pyruvate + CO_2_ + NADPH, Oxaloacetate <=> pyruvate + CO_2_	*mae*B
ING2E5A_3022	2.7.9.1	Pyruvate, phosphate dikinase	ATP + pyruvate + phosphate <=> AMP + phosphoenolpyruvate + diphosphate	*ppd*K

**Table 3 microorganisms-08-02024-t003:** Most actively transcribed genes for enzymes involved in carbohydrate or amino acid utilization in *P. mucosa* which were ranked by TPM values from highest to lowest as deduced from publicly available metatranscriptome dataset (accession numbers: ERR3010911, ERR3010910, ERR3010909, ERR3010916).

Ranking	GenID	Gene Length	Putative Gene Product	Gene	EC Number	Transcriptome Reads Mapped	Normalized Number of Transcripts (TPM)
7	ING2E5A_1308	1191	Aspartate aminotransferase	*ast*	2.6.1.1	739	21,591
8	ING2E5A_1484	1179	6-phosphogluconolactonase	*pgls*	3.1.1.31	178	5255
21	ING2E5A_1407	1500	Propionyl-CoA:succinate CoA transferase	*scp*C	2.8.3.-	186	4328
26	ING2E5A_0536	720	Peptidase E	*pep*E	3.4.13.21	76	3688
32	ING2E5A_3029	1191	Aspartate aminotransferase	*ast*	2.6.1.1	113	3315
37	ING2E5A_0388	657	Uracil phosphoribosyltransferase	*uprt*	2.4.2.9	57	3049
43	ING2E5A_2638	1521	Periplasmic serine endoprotease	*deg*P	3.4.21.107	121	2774
50	ING2E5A_0405	1452	6-phosphogluconate dehydrogenase, decarboxylating	*gnd*A	1.1.1.44	97	2344
85	ING2E5A_0914	3792	TonB-dependent outer-membrane transporter	*sus*C	n.d.	205	1887
94	ING2E5A_0692	1014	ATP-dependent 6-phosphofructokinase	*pfk*	2.7.1.11	50	1725

n.d. = not determined.

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
