# Peer review of "The Role of Petrimonas mucosa ING2-E5AT in Mesophilic Biogas Reactor Systems as Deduced from Multiomics Analyses"

_microorganisms, 2020, doi:10.3390/microorganisms8122024_

Round 1

Reviewer 1 Report

The paper is a valuable contribution to AD microbiology and nicely combines genome characterization of a single isolate and multi-omics analyses of public data to gain more information on the specific role of a key species in AD processes. In general, the paper is well written but requires some improvements regarding strcuture, language and the quality of figures. It seems that the single chapters have been written by different authors and put together without a final consistency check (see details below). Language check by a native speaker or an academic editing service is recommended, e.g. to improve correct syntax and tense, use of commas and articles, which vs. that ...

Introduction:

L 54: Energy cannot be generated, only energy carriers. I suggest to rephrase this.

L 55: microbial process

L 57: Which changes and disturbances do you mean? Without more specific information, this sentence is vague and simply a platitude.

L 59: Is foaming really a cause of process inhibition or rather an effect or symptom?

L 63: For such a general statement, a review article or book chapter should be cited rather than an arbitrarily selected single research paper that even has a different scope.

L 63-65: "supports" seems to be the wrong word here. Apart from the odd wording, this sentence confuses cause with effect. Ammonia leads to the inhibition of methanogens and probably also of syntrophic VFA oxidizers, which results in VFA accumulation and consequently further inhibition of methanogens. However, the primary inhibitory effect is exerted by ammonia itself and not by pH decrease - high ammonia concentration even counteracts low pH due to its buffer capacity. I suggest to rephrase this sentence and also to connect it better to the subsequent paragraph, which starts with a very general statement that does not specifically refer to AD microbiomes. Correct "in summary resulting in"

L 74 and elsewhere: correct "Dysgonomonadaceae" (not "Dysgomonadaceae").

L 87: Impact of what on what? Rephrase this sentence.

L 91: genera, not species

L 92: warning indicators for what?

L 94: Replace "which" by "that"

Methods section:

L 116: write the unit directly after the number (1.7 g/L sodium ...)

L 118: Why was an additional purification step needed after using a commercial DNA extraction kit for a pure culture? How was the DNA quality and concentration assessed to be sufficient for PacBio sequencing?

L 131: Ref. 9 refers to metagenome data. Please explain how Illumina reads from were selected to be specific for strain ING2-E5A. I cannot find a Petrimonas MAG in paper [9].

L 144: Give a reference or URL for KEGG.

L 145: Why not using a common format for 10^-100? (using superscription of the exponent)?

L 159: Give a reference or URL for ENA.

L 164: What was the source of the datasets?

L 178: "counts were split with each ..." - this is unclear, rephrase this sentence.

L 180: What does this equation mean? An explanation referring to it is missing. And what is the meaning of the lines framing the equation on two of four sides?

L 182: What is li? In the equation, the i is subscribed.

L 192: What is the original study? Give the reference or accession number of the metagenome dataset.

L 195-196: "if they shared ... when they shared ..."? Wrong syntax, rephrase. Instead of what?

L 198: Do you mean ENA (which is the sequence database of EMBL-EBI)? Give the URL to facilitate finding the genome sequence.

Results and Discussion:

General comments: Do not repeat methods and their references in the results section, but describe all methods used in Section 2. Make sure that all genus and species names as well as gene names are written in italics. Check numbering of the sections (there are two sections 3.4).

Section 3.1: The title should be "General features of the ... genome" - this section is not about features of the strain but merely of its genome. Move methods descriptions to section 2. Avoid redundancies between text, Table 1 and Figure 1 (for instance, the genome size is mentioned three times!). The text says "more than 3000 coding sequences", but according to Table 1 there are exactly 3000 CDS. Figure 1 is too small, enlarge the genome map.

L 233: A separate or a distinct cluster? The two attributes are somewhat redundant.

L 239: The type species P. sulfuriphila is missing in the tree. I assume this is due to the lack of a genome sequence, but this should be mentioned here.

L 240: It is not true that P. mucosa is the only species described for this genus. P. sulfuriphila is also a validly described species of this genus, as stated in the preceding sentence.

L 244-245: Check correct tense. "currently" and past tense ("were described") exclude each other.

L 246 and 249: Introduce abbreviations only when they are used again at least once (UASB and UASS).

L 244-260: Somehow these two paragraphs are not related to the scope of this paper. The paper describes Petrimonas mucosa - why are other genera discussed here? I suggest to delete this part to streamline the paper and focus on the actual results.

Figure 2 is too small and the resolution is too low. Enlarge the figure, improve the resolution and use a larger font for the tree. Figure caption: "Phylogenetic tree of the order ..." (not: "for the order").

Section 3.3: Shorten the title by removing the part "involved in ..."

L 275: Reference for the CAZypedia Consortium?

Figure 3 is again too small, enlarge it. Why is this diagram shown as a ring implying a genome map?

Section 3.4 (L 307): Remove the species/strain name in the title and start with "Genes encoding ..." to make this title consisten with that of section 3.3.

L 311-313: This sentence contradicts the preceding one. "were considered, in combination ..., as an ..." sounds awkward. Rephrase this like: which were suggested to be

L 329: Why host glycoprotiens? Which host?

L 346: "carefully be assumed"? Rephrase. Just "can be assumed" is sufficient.

Section 3.4 (L 354): The title is too specific and does not reflect the content of this section. Rephrase and shorten it, e.g.: Genes encoding enzymes of the central fermentation metabolism

L 366: Replace "within". The correct preposition could be "by", "through", "via", "by means of" ...

L 373-375: Reference for this statement?

L 375: Simply carbon dioxide or CO2 is sufficient; no need to explain chemical formulas

L 380: What is missing to make the pathway complete?

L 385: Replace "similar to" by "just like" or "just as" or "as well as". It belongs to the same order, thus, the affiliation is not similar but identical.

L 397: Correct subscriptions in O2 and H2, or use the words oxygen and hydrogen. "has an unconfirmed role" sounds awkward, rephrase.

L 398: several subunits? How many? Give a number.

L 406-408: Remove this explanation here in the footnote. Such explanations should be given in the main text only.

L 427: Replace "amount" by "number".

L 431: Reference for Brandt and colleagues?

L 433: Explain abbreviations at the first use (BGP).

L 442: Which niche? "a certain niche" is too vague - could you specify it?

L 447: Replace "widely spread" by "widespread"

L 449: not only lignocellulosic biomass, but also proteins (including turnover of microbial biomass, which could explain their increase during process disturbances)

Figure 5: Enlarge the figure and use thousand delimiters in the read numbers. Explain the meaning of GCA_... Remove "at the NCBI" in the figure caption. The accession numbers refer to all three public sequence databases (not only NCBI).

Section 3.6. Title. Why only sugar fermentation? The results show also a metabolism based on peptide fermentation. Keep the title short and do not exclude results.

L 464: What is meant by "one to five"?

L 468: "comprise" and "those that did" is not the same tense - make the two sentences consistent as the latter refers to the first.

L 469: Remove the hyphen in "DNA reads".

L 480 and elsewhere: Write "datasets" as one word to be consistent with the other sections of the manuscript (check the whole manuscript for such inconsistencies).

L 488: on average

Table 3: Remove "at the NCBI" in the title.

L 525: "technical and chemical process parameters"? Do you mean operational parameters (set by the operator, e.g. HRT, OLR, temperature, ...) vs. process parameters (resulting from the operational conditions, e.g. pH, VFA level, gas amount and composition, ...)?

L 533-534: How can peptides be detected in biogas samples? And why "on average" when detected in all samples?

Figure 6 is not readable. Figure caption: What is meant by "against the same database"? Which database? CSTR means "continuous strirred tank reactor", not "continuously" (a CSTR is not continuously stirred, but fed in continuous mode).

Conclusion:

L 560: "approaches ... as tool" - decide for plural or singular.

L 562-565: Rephrase this sentence, the syntax makes no sense: "the genome sequence ... was accompanied by the analysis of a polyphasic approach comprising ...datasets" ???

L 572: "encodes ... genes" makes no sense (delete "genes"). The second part of the sentence ("mostly ...") refers to genes and needs to be rephrased (could be a separate sentence).

L 586: Again, which niche? The reader would expect some specific information. Of course, any organism occupies a certain niche, this is a platitude.

L 587 What is meant by "competitive bacteria"?

L 591: "biogas plant community members" sounds awkward. The communities are in the reactor, not in the plant. A more common term could be "AD community members" or similar.

L 593-594: Really? How should this work? Such a claim needs to be explained or simply omitted.

References: Check format and style (should be consistent and in accordance with the journal style) as well as completeness (many references are incomplete, e.g. page numbers are missing). Check correct format of taxonomic names in the references (genus and species names in italics, sp. nov. and gen. noc etc. in lower case).

Author Response

We thank Reviewer #1 for his positive evaluation of our manuscript and underlining the importance of multi-omics analyses in studies like ours. Please find our point-by-point response to the reviewer’s comments attached to this submission.

Manuscript ID: microorganisms-1017890

Point-to-Point Response to Reviewer Reports

Reviewer #1: The paper is a valuable contribution to AD microbiology and nicely combines genome characterization of a single isolate and multi-omics analyses of public data to gain more information on the specific role of a key species in AD processes. In general, the paper is well written but requires some improvements regarding structure, language and the quality of figures. It seems that the single chapters have been written by different authors and put together without a final consistency check (see details below). Language check by a native speaker or an academic editing service is recommended, e.g. to improve correct syntax and tense, use of commas and articles, which vs. that ...

Response 1:  We thank Reviewer #1 for his positive evaluation of our manuscript and underlining the importance of multi-omics analyses in studies like ours.

Introduction:

Reviewer #1: L 54: Energy cannot be generated, only energy carriers. I suggest to rephrase this.

Response 2:  The sentence has been rephrased and now reads (lines 53-54): ‘Anaerobic digestion (AD) is commonly applied for treatment of organic wastes in order to achieve reduction of waste combined with parallel recovery of bioenergy.‘

Reviewer #1: L 55: microbial process

Response 3:  It has been corrected.

Reviewer #1: L 57: Which changes and disturbances do you mean? Without more specific information, this sentence is vague and simply a platitude.

Response 4:  The sentence has been rephrased and now reads (lines 57-58): ‘The microbial biogas network is complex and intertwined and therefore susceptible to environmental changes.’

Reviewer #1: L 59: Is foaming really a cause of process inhibition or rather an effect or symptom?

Response 5: The formation of foam in a biogas fermenter can have different causes. These causes can be of biological or mechanical nature, but they can also be based on unsuitable process management or substrate selection. When a strong foam event has occurred in the fermenter, this can have serious consequences for technology and plant safety e. g. contamination of the entire gas supply system, crust formation in the gas area, damage of the fermenter roof or the gas storage foil caused by the pressure that had build-up. Technical problems frequently lead to the disturbance of the biocenosis, which ultimately can lead to a drop in methane production.

Reviewer #1: L 63: For such a general statement, a review article or book chapter should be cited rather than an arbitrarily selected single research paper that even has a different scope.

Response 6: A review about ammonia inhibition in anaerobic digestion was cited at the corresponding position: 

Yenigün O. and Demirel B. Ammonia inhibition in anaerobic digestion: A review. Process Biochemistry 2013, 48(5-6): 901-911.

Reviewer #1: L 63-65: "supports" seems to be the wrong word here. Apart from the odd wording, this sentence confuses cause with effect. Ammonia leads to the inhibition of methanogens and probably also of syntrophic VFA oxidizers, which results in VFA accumulation and consequently further inhibition of methanogens. However, the primary inhibitory effect is exerted by ammonia itself and not by pH decrease - high ammonia concentration even counteracts low pH due to its buffer capacity. I suggest to rephrase this sentence and also to connect it better to the subsequent paragraph, which starts with a very general statement that does not specifically refer to AD microbiomes. Correct "in summary resulting in"

Response 7: The sentence has been modified and now reads: ‘Due to inhibition of syntrophic volatile fatty acid (VFA) oxidizers, high ammonia concentrations result in VFA accumulation which subsequently inhibits methanogenesis. Moreover, ammonia also directly inhibits methanogens (Rajagopal et al., 2013) (line 63).’

Rajinikanth Rajagopal, Daniel I.Massé, Gursharan Singh. A critical review on inhibition of anaerobic digestion process by excess ammonia. Bioresource Technology 143: 632-641.

Reviewer #1: L 74 and elsewhere: correct "Dysgonomonadaceae" (not "Dysgomonadaceae").

Response 8: Thank you. This mistake has been corrected throughout the text.

Reviewer #1: L 87: Impact of what on what? Rephrase this sentence.

Response 9: The sentence has been rephrased as follows (line 87): ‘The impact of Dysgonomonadaceae members on biomass degradation in AD reactor systems operating at high OLR or increased nitrogen/ammonia levels caused by proteinaceous substrates has been demonstrated recently [9, 10].‘

Reviewer #1: L 91: genera, not species

Response 10: It has been corrected.

Reviewer #1: L 92: warning indicators for what?

Response 11: The sentence has been rephrased and now reads (line 91): ‘These observations led to the assumption that these genera may serve as microbial early warning indicators for an unbalanced process . They are therefore regarded as potential marker microorganisms featuring outstanding performance under stress conditions.’

Reviewer #1: L 94: Replace "which" by "that"

Response 12: It has been corrected (line 95).

Methods section:

Reviewer #1: L 116: write the unit directly after the number (1.7 g/L sodium ...)

Response 13:  Done (line 140).

Reviewer #1: L 118: Why was an additional purification step needed after using a commercial DNA extraction kit for a pure culture? How was the DNA quality and concentration assessed to be sufficient for PacBio sequencing?

Response 14: Sample purification is critical for reliable sequence data, and the primary requirement for a successful sequencing approach is a nucleic acid template that is of high quality and purity. P. mucosa DNA purity and quality was determined via the ratios of A260 to A280 and A260 to A230 using the NanoDrop 2000 device. Approximately 8 µg high-molecular-weight genomic DNA with A260/280 and A260/230 values greater than 1.8 was required for SMRTbellTM template library construction. Unfortunately, after utilization of the Gentra Puregene Yeast/Bact. Kit (Qiagen, Hilden, Germany) for DNA extraction, the required DNA purity was not achieved. Thus, an additional purification step with the NucleoSpin® gDNA Clean-up kit was carried out. This information is now included in the material and methods section in lines 142-145.

Reviewer #1: L 131: Ref. 9 refers to metagenome data. Please explain how Illumina reads from were selected to be specific for strain ING2-E5A. I cannot find a Petrimonas MAG in paper [9].

Response 15: We thank Reviewer 1 for drawing our attention to this issue. The reference number 9 was wrongly cited here. The correct reference is the following one:

Maus et al. Genomics and prevalence of bacterial and archaeal isolates from biogas-producing microbiomes. Biotechnology for Biofuels 2017, 10(1): 264.

This reference has now been included in the text in line 157.

Reviewer #1: L 144: Give a reference or URL for KEGG.

Response 16: Done (line 171).

Reviewer #1: L 145: Why not using a common format for 10^-100? (using superscription of the exponent)?

Response 17: Corrected accordingly (line 174).

Reviewer #1: L 159: Give a reference or URL for ENA.

Response 18: Done (line 181).

Reviewer #1: L 164: What was the source of the datasets?

Response 19: The sentence has been rephrased and the source of the datasets was introduced (line 183): ‘Metatranscriptome datasets pertaining to samples from different industrial- and laboratory-scale anaerobic digesters, detailed in Supporting Information Table 3, were re-analyzed to quantify transcripts of Petrimonas mucosa.‘

Reviewer #1: L 178: "counts were split with each ..." - this is unclear, rephrase this sentence.

Response 20: The sentence has been rephrased and now reads (lines 206-210): Individual contigs may overlap multiple coding sequences of the genome. In such cases, the read counts that were computed for a contig were split up between all coding sequences which the contig is overlapping. A fraction X_i of total contig-read-counts X is assigned to each coding sequence i. X_i is determined by dividing the length of the sequence l_i through the sum of the length of all sequences j which are overlapping the contig:"

Reviewer #1: L 180: What does this equation mean? An explanation referring to it is missing. And what is the meaning of the lines framing the equation on two of four sides?

Response 21: The equation details how reads were counted in cases where a contig is overlapping multiple coding sequences. The text passage above the formula has been adjusted to provide a clearer explanation. The framing lines are merely an artifact of formatting because we could only include the formula as an image.

Reviewer #1: L 182: What is li? In the equation, the i is subscribed.

Response 22: The i should always be subscribed. l_i denotes the length of coding sequence i.

Reviewer #1: L 192: What is the original study? Give the reference or accession number of the metagenome dataset.

Response 23: The references for the metagenome data are now introduced in line 229:

  1. Rademacher A, Zakrzewski M, Schluter A, Schonberg M, Szczepanowski R, Goesmann A, Puhler A, Klocke M. Characterization of microbial biofilms in a thermophilic biogas system by high-throughput metagenome sequencing. FEMS Microbiol Ecol. 2012;79:785–99.
  2. Hanreich A, Schimpf U, Zakrzewski M, Schluter A, Benndorf D, Heyer R, Rapp E, Puhler A, Reichl U, Klocke M. Metagenome and metaproteome analyses of microbial communities in mesophilic biogas-producing anaerobic batch fermentations indicate concerted plant carbohydrate degradation. Syst Appl Microbiol. 2013;36:330–8.
  3. Stolze Y, Bremges A, Rumming M, Henke C, Maus I, Puhler A, Sczyrba A, Schluter A. Identification and genome reconstruction of abundant distinct taxa in microbiomes from one thermophilic and three mesophilic production-scale biogas plants. Biotechnol Biofuels. 2016;9:156.
  4. Schlüter A, Bekel T, Diaz NN, Dondrup M, Eichenlaub R, Gartemann KH, Krahn I, Krause L, Kromeke H, Kruse O, et al. The metagenome of a biogas-producing microbial community of a production-scale biogas plant fermenter analysed by the 454-pyrosequencing technology. J Biotechnol. 2008;136:77–90.

Reviewer #1: L 195-196: "if they shared ... when they shared ..."? Wrong syntax, rephrase. Instead of what?

Response 24: The sentence has been rephrased and now reads (lines 229-231): ‘The remaining settings were used as described previously by Heyer and colleagues [56] with the minor change that redundant homologous proteins were grouped when they shared the same peptide.’

Reviewer #1: L 198: Do you mean ENA (which is the sequence database of EMBL-EBI)? Give the URL to facilitate finding the genome sequence.

Response 23: Done (line 234).

Results and Discussion:

Reviewer #1: General comments: Do not repeat methods and their references in the results section, but describe all methods used in Section 2. Make sure that all genus and species names as well as gene names are written in italics. Check numbering of the sections (there are two sections 3.4).

Response 24: The redundancy was removed within the entire chapter 3 (from line 236). According to guidelines for authors provided by the journal Microorganisms, italics must be used for the genus and species only when using Latin names of organisms:

https://www.mdpi.com/authors/layout#_bookmark15

Reviewer #1: Section 3.1: The title should be "General features of the ... genome" - this section is not about features of the strain but merely of its genome. Move methods descriptions to section 2. Avoid redundancies between text, Table 1 and Figure 1 (for instance, the genome size is mentioned three times!). The text says "more than 3000 coding sequences", but according to Table 1 there are exactly 3000 CDS. Figure 1 is too small, enlarge the genome map.

Response 25: The title and the text of the chapter 3.1 has been adjusted accordingly (from line 237). We also provided high resolution figures for the manuscript as separate files. These are large enough to see all text elements and numbers. Obviously, these files were not passed on to the reviewers. I am sorry that the Reviewer had not got access to these high-resolution images.

Reviewer #1: L 233: A separate or a distinct cluster? The two attributes are somewhat redundant.

Response 26: Suggestion gladly accepted. The word ‘distinct’ has been removed.

Reviewer #1: L 239: The type species P. sulfuriphila is missing in the tree. I assume this is due to the lack of a genome sequence, but this should be mentioned here.

Response 27: Suggestion gladly accepted. The Reviewer is right, the genome of P. sulfuriphila has not yet been sequenced.

Reviewer #1: L 240: It is not true that P. mucosa is the only species described for this genus. P. sulfuriphila is also a validly described species of this genus, as stated in the preceding sentence.

Response 28: Thank you for this comment. The sentence has been rephrased and now reads (lines 284-286): ‘So far, P. mucosa and P. sulfuriphila were the only species described for this genus validly.’

Reviewer #1: L 244-245: Check correct tense. "currently" and past tense ("were described") exclude each other.

Response 29: The entire paragraph was removed as suggested later (line 280).

Reviewer #2: L 246 and 249: Introduce abbreviations only when they are used again at least once (UASB and UASS).

Response 30: The entire paragraph was removed as suggested in the next comment.

Reviewer #1: L 244-260: Somehow these two paragraphs are not related to the scope of this paper. The paper describes Petrimonas mucosa - why are other genera discussed here? I suggest to delete this part to streamline the paper and focus on the actual results.

Response 31: We followed the Reviewer’s suggestion and removed these two paragraphs from the manuscript (line 280).

Reviewer #1: Figure 2 is too small and the resolution is too low. Enlarge the figure, improve the resolution and use a larger font for the tree. Figure caption: "Phylogenetic tree of the order ..." (not: "for the order").

Response 32: Regarding the figure 2, please read our comment above. To respond to the reviewer's criticism, we enlarged the font in the figure 2 (please see new Figure 2 provided in the manuscript). The mistake has been corrected (line 281).

Reviewer #1: Section 3.2: Shorten the title by removing the part "involved in ..."

Response 33: The title has been shortened (line 332).

Reviewer #1: L 275: Reference for the CAZypedia Consortium?

Response 34: Thank you for this hint. The reference is now introduced in line 339:

  1. The CAZypedia Consortium. Ten years of CAZypedia: a living encyclopedia of carbohydrate-active enzymes. Glycobiology 2018, Volume 28(1): 3-8.

Reviewer #1: Figure 3 is again too small, enlarge it. Why is this diagram shown as a ring implying a genome map?

Response 35: Regarding the figure 3, please read our comment above. The diagram was shown as a ring for optical reasons. In our opinion, this picture looks nice and at the same time easy to understand. All numbers will be legible, when the graphic will be enlarged in the printed version of the article. The corresponding (original) vector graphic was attached to this submission.

Reviewer #1: Section 3.3 (L 307): Remove the species/strain name in the title and start with "Genes encoding ..." to make this title consisten with that of section 3.3.

Response 36: The title has been adjusted as suggested (line 373).

Reviewer #1: L 311-313: This sentence contradicts the preceding one. "were considered, in combination ..., as an ..." sounds awkward. Rephrase this like: which were suggested to be.

Response 37: Has been corrected accordingly.

Reviewer #1: L 329: Why host glycoprotiens? Which host?

Response 38: The word ‘host’ has been removed (line 398).

Reviewer #1: L 346: "carefully be assumed"? Rephrase. Just "can be assumed" is sufficient.

Response 39: Done (line 415).

Reviewer #1: Section 3.4 (L 354): The title is too specific and does not reflect the content of this section. Rephrase and shorten it, e.g.: Genes encoding enzymes of the central fermentation metabolism

Response 40: The suggestion of the Reviewer for the title of the chapter 3.4 has been accepted (line 423).

Reviewer #1:  L 366: Replace "within". The correct preposition could be "by", "through", "via", "by means of" ...

Response 41: The preposition ‚within’ has been replaced by ‚via‘ (line 424).

Reviewer #1:  L 373-375: Reference for this statement?

Response 42: According to ExPASy (https://enzyme.expasy.org), acetate kinase (EC 2.7.2.1) can also use propanoate as a substrate. Acetate can be converted into the intermediate acetyl- CoA by coupling the reaction of acetate kinase with phosphate acetyltransferase (EC 2.3.1.8) activity. Both, this enzyme and acetate kinase also are important in the production of propanoate. The reference for ExPASy (https://enzyme.expasy.org) is now given in the text (line 442).

Reviewer #1:  L 375: Simply carbon dioxide or CO2 is sufficient; no need to explain chemical formulas.

Response 43: The formula for carbon dioxide has been deleted.

Reviewer #1:  L 380: What is missing to make the pathway complete?

Response 44: Genes for the enzymes involved in conversion of 3-hydroxybutanoyl-CoA to crotonoyl-CoA (or vice versa) and crotonoyl-CoA to butanoyl-CoA could not be identified in the P. mucosa ING2-E5AT genome. This information is now also given in the text (line 452).

Reviewer #1:  L 385: Replace "similar to" by "just like" or "just as" or "as well as". It belongs to the same order, thus, the affiliation is not similar but identical.

Response 45: The word ‚similar‘ has been replaced by ‚as well as‘ (line 447).

Reviewer #1:  L 397: Correct subscriptions in O2 and H2, or use the words oxygen and hydrogen. "has an unconfirmed role" sounds awkward, rephrase.

Response 46: The terms ‚oxygen‘ and ‚hydrogen‘ are now used in the text. The sentence has been rephrased and now reads: ‚ … and its role remains unclear so far.‘ (line 470).

Reviewer #1:  L 398: several subunits? How many? Give a number.

Response 47: The heterotetrameric complex of Hnd is composed of HndA, HndB, HndC and HndD subunits. This information is now provided in the text (line 471).

Reviewer #1:  L 406-408: Remove this explanation here in the footnote. Such explanations should be given in the main text only.

Response 48: The explanatory footnote has been removed. Corresponding information is given in the text (lines 442-445).

Reviewer #1:  L 427: Replace "amount" by "number".

Response 49: Done.

Reviewer #1:  L 431: Reference for Brandt and colleagues?

Response 50: The reference for Brandt et al. is reference No. [83] (line 514).

Reviewer #1:  L 433: Explain abbreviations at the first use (BGP).

Response 51: Done.

Reviewer #1:  L 442: Which niche? "a certain niche" is too vague - could you specify it?

Response 52: Since the niche that is occupied by P. mucosa cannot be precisely defined, the term ‚niche‘ is avoided. Therefore, the sentence has been rephrased as follows:

‚Most probably, P. mucosa was able to cope with corresponding stress conditions and displaced other competing bacteria that are less stress resistant (line 524).‘

Reviewer #1:  L 447: Replace "widely spread" by "widespread"

Response 53: Done.

Reviewer #1:  L 449: not only lignocellulosic biomass, but also proteins (including turnover of microbial biomass, which could explain their increase during process disturbances)

Response 54: We followed the Reviewer’s suggestion and rephrased the sentence as follows (line 542): ‘Under these conditions, they may play an important role in conversion of lignocellulosic biomass and also proteins for subsequent methane production.’

Reviewer #1:  Figure 5: Enlarge the figure and use thousand delimiters in the read numbers. Explain the meaning of GCA_... Remove "at the NCBI" in the figure caption. The accession numbers refer to all three public sequence databases (not only NCBI).

Response 55: The figure 5 was updated accordingly. All abbreviations were explained (line 549).

Reviewer #1:  Section 3.6. Title. Why only sugar fermentation? The results show also a metabolism based on peptide fermentation. Keep the title short and do not exclude results.

Response 56: The title has been shortened (line 551).

Reviewer #1:  L 464: What is meant by "one to five"?

Response 57: The sentence has been rewritten and now reads (line 562):Unfortunately, no metatranscriptome data exist for the metagenomes presented in Figure 5 referring to the fragment recruitment analyses (see above).’

Reviewer #1:  L 468: "comprise" and "those that did" is not the same tense - make the two sentences consistent as the latter refers to the first.

Response 58: The sentence has been rewritten and now reads (line 566): For metagenome data for which also metatranscriptome data was available, only fractions of the P. mucosa genome could be assembled from the DNA reads.

Reviewer #1:  L 469: Remove the hyphen in "DNA reads".

Response 59: Done.

Reviewer #1:  L 480 and elsewhere: Write "datasets" as one word to be consistent with the other sections of the manuscript (check the whole manuscript for such inconsistencies).

Response 60: Done.

Reviewer #1:  L 488: on average

Response 61: The sentence was slightly modified and now reads: ‘Based on abundances of spectral counts, peptides originating from P. mucosa proteins after tryptic digestion were detected in all biogas samples analyzed, showing abundance values in the range of 0.2% to 1.4% of all spectra identified (line 636).’

Reviewer #1:  Table 3: Remove "at the NCBI" in the title.

Response 62: Done.

Reviewer #1:  L 525: "technical and chemical process parameters"? Do you mean operational parameters (set by the operator, e.g. HRT, OLR, temperature, ...) vs. process parameters (resulting from the operational conditions, e.g. pH, VFA level, gas amount and composition, ...)?

Response 63: Thank you for this hint. The sentence was slightly modified and now reads: ‘Further details on operational and process parameters of the investigated BGPs as well as fed substrates are summarized in Figure 6 (line 628).’

Reviewer #1:  L 533-534: How can peptides be detected in biogas samples? And why "on average" when detected in all samples?

Response 64: The proteomics/metaproteomics workflow includes digestion of proteins by trypsin resulting in the release of peptides which subsequently are subjected to MALDI-ToF Mass Spectrometry to determine their masses.

The corresponding sentence in the text now reads: ‚Based on abundances of spectral counts, peptides originating from P. mucosa proteins after tryptic digestion were detected in all biogas samples analyzed, showing abundance values in the range of 0.2% to 1.4% of all spectra identified (line 636).‘

Reviewer #1:  Figure 6 is not readable. Figure caption: What is meant by "against the same database"? Which database? CSTR means "continuous strirred tank reactor", not "continuously" (a CSTR is not continuously stirred, but fed in continuous mode).

Response 65: Regarding the picture quality, please read our Response 25. Figure legend has been adjusted and the corresponding sentence reads as follows: ‘The measurements were searched with X!tandem and OMSSA against the protein database derived from original data extended with the sequences of Petrimonas mucosa (line 659).’The meaning of CSTR has been corrected accordingly (line 652 ).

Conclusion:

Reviewer #1:  L 560: "approaches ... as tool" - decide for plural or singular.

Response 66: The sentence has been rephrased as follows: ’Integrated omics analyses are useful for a detailed characterization of both, entire microbiomes and individual microbial species enabling new ways to access the microbiological basis for AD (line 668).’

Reviewer #1:  L 562-565: Rephrase this sentence, the syntax makes no sense: "the genome sequence ... was accompanied by the analysis of a polyphasic approach comprising ...datasets" ???

Response 67: The sentence has been rephrased as follows: ‘In this study, the genome sequence of P. mucosa ING2-E5AT isolated from a mesophilic laboratory-scale reactor was analyzed in detail including genome-based metabolic reconstruction. In addition, a polyphasic approach exploiting publicly available metagenome, metatranscriptome, and metaproteome datasets complemented the genome interpretation of the strain (line 669).’

Reviewer #1:  L 572: "encodes ... genes" makes no sense (delete "genes"). The second part of the sentence ("mostly ...") refers to genes and needs to be rephrased (could be a separate sentence).

Response 68: The sentence has been rephrased as follows: ‚Insights into the strain’s genetic repertoire and metabolic potential revealed that P. mucosa ING2-E5AT encodes a diverse set of glycosyl-hydrolyses (GH) involved in carbohydrate metabolism as well as pectin degradation. Most of the corresponding genes are organized in Polysaccharide-Utilization-Loci (PUL) also comprising tandem susCD-like genes for a TonB-dependent outer-membrane transporter and a cell surface glycan-binding protein (line 679).‘

Reviewer #1:  L 586: Again, which niche? The reader would expect some specific information. Of course, any organism occupies a certain niche, this is a platitude.

Response 69: Since the niche that is occupied by P. mucosa cannot be precisely defined, the term ‚niche‘ is avoided. Therefore, the sentence has been rephrased as follows: ‚Most probably, P. mucosa was able to cope with corresponding stress conditions and displaced other competing bacteria that are less stress resistant (line 716).‘

Reviewer #1:  L 587 What is meant by "competitive bacteria"?

Response 70: The corresponding sentence has been rephrased (see above L. 716).

Reviewer #1:  L 591: "biogas plant community members" sounds awkward. The communities are in the reactor, not in the plant. A more common term could be "AD community members" or similar.

Response 71: The term ‚AD community members‘ suggested by the Reviewer is now used in the text (line 722).

Reviewer #1:  L 593-594: Really? How should this work? Such a claim needs to be explained or simply omitted.

Response 72: The corresponding sentence has been rephrased as follows: ‚The knowledge gained from multi-omics analyses will provide the basis for application of the strain in demanding biotechnological processes. P. mucosa may also be used as indicator species for unbalanced AD processes in the future. However, further information on the performance of the strain in different AD microbiomes exposed to unfavorable process conditions is needed to precisely determine its functional behavior (line 722).’

Reviewer #1:  References: Check format and style (should be consistent and in accordance with the journal style) as well as completeness (many references are incomplete, e.g. page numbers are missing). Check correct format of taxonomic names in the references (genus and species names in italics, sp. nov. and gen. noc etc. in lower case).

Response 72: Format and style of references has been checked. Likewise, the correct format of taxonomic names in the references has been checked.

Reviewer 2 Report

Petrimonas mucosa ING2-E5AT was isolated from a mesophilic biogas reactor systems in a previous study (reference24). In this study, the authors sequenced the PacBio RSII and Illumina MiSeq sequencers for metagenomics. The authors make a good job from the enzymatic and genetic point of view.   It is not clear how to get RNA data. RNA extraction lacks.   line32 und --> and   result or table 1 show the completeness and AAI/ANI.   conclusion Give a strong and specified conclusion for this research.   supporting data Explain supporting information 4 in supplemental Materials (line 595-600).

Author Response

We thank Reviewer #2 for his positive evaluation of our manuscript. Please find our point-by-point response to the reviewer’s comments attached to this submission.

Manuscript ID: microorganisms-1017890

Point-to-Point Response to Reviewer Reports

Reviewer #2: Petrimonas mucosa ING2-E5AT was isolated from a mesophilic biogas reactor systems in a previous study (reference24). In this study, the authors sequenced the PacBio RSII and Illumina MiSeq sequencers for metagenomics. The authors make a good job from the enzymatic and genetic point of view. It is not clear how to get RNA data. RNA extraction lacks.  

Response 1:  We thank Reviewer #2 for his mostly positive evaluation of our manuscript. Metatranscriptome sequence data were obtained from publicly available resources, namely the NCBI sequence database. References for the datasets used in this study were summarized and listed in Supplementary Table 3. Likewise, original RNA-extraction protocols are given in the original publications (also listed in Supplementary Table 3 if available) associated with the datasets used in this study.

Reviewer #2: line32 und --> and  

Response 2:  The mistake has been corrected.

Reviewer #2: result or table 1: show the completeness and AAI/ANI.

Response 3:  In this study, we are presenting the completely finished genome sequence for P. mucosa ING2-E5AT (100% completeness). The phylogenetic relation of the strain is addressed in chapter 3.2 and in Fig. 2. For details, please also refer to the Figure legend of Fig. 2.

Reviewer #2: Conclusion: Give a strong and specified conclusion for this research.  

Response 4:  Key sentences of the conclusion chapter were rephrased and modified. We hope that the conclusion drawn from this research has now clearly been presented. Likewise, the main findings of this paper are summarized in the abstract.

Reviewer #2: supporting data: Explain supporting information 4 in supplemental Materials (line 595-600).

Response 5: Supporting information No. 4 has now been described in the Supplementary Materials paragraph (line 734).
